# LIFELONG LEARNING WITH OUTPUT KERNELS

## ABSTRACT

Lifelong multitask learning poses considerable challenges in terms of effectiveness (minimizing prediction errors for all tasks) and overall computational tractability for real-time performance. This paper addresses continuous lifelong multitask learning by jointly re-estimating the inter-task relations (*output* kernel) and the per-task model parameters at each round, assuming data arrives in a streaming fashion. A new algorithm called *Online Output Kernel Learning Algorithm* (OOKLA) for lifelong learning setting is proposed. To avoid the memory explosion, a robust budget-limited versions of the proposed algorithm is introduced, which efficiently utilizes the relationships between the tasks to bound the total number of representative examples in the support set. In addition, a two-stage budgeted scheme for efficiently tackling the task-specific budget constraints in lifelong learning is proposed. Empirical results over three datasets indicate superior AUC performance for OOKLA and its budget-limited cousins over strong baselines.

## 1 INTRODUCTION

Instead of learning individual models, learning from multiple tasks leverages the relationships among tasks to jointly build better models for each task and thereby improve the transfer of relevant knowledge between the tasks, especially from information-rich tasks to information-poor ones. Unlike traditional multitask learning, where the tasks are presented simultaneously and an entire training set is available to the learner (Caruana (1998)), in lifelong learning the tasks arrives sequentially (Thrun (1996)). This paper considers a continuous lifelong learning setting in which both the tasks and the examples of the tasks arrive in an online fashion, without any predetermined order.

Following the online setting, particularly from Saha et al. (2011); Cavallanti et al. (2010), at each round $t$, the learner *receives* an example from a task, along with the task identifier and *predicts* the output label for the example. Subsequently, the learner receives the true label and *updates* the model(s) as necessary. This process is repeated as we receive additional data from the same or different tasks. Our approach follows an error-driven update rule in which the model for a given task is updated only when the prediction for that task is in error.

Lifelong learning poses considerable challenges in terms of effectiveness (minimizing prediction errors for all tasks) and overall computational tractability for real-time performance. A lifelong learning agent must provide an efficient way to learn new tasks faster by utilizing the knowledge learned from the previous tasks and also not forgetting or significantly degrading performance on the old tasks. The goal of a lifelong learner is to minimize errors as compared to the full ideal hindsight learner, which has access to all the training data and no bounds on memory or computation.

This paper addresses lifelong multitask learning by jointly re-estimating the inter-task relations from the data and the per-task model parameters at each round, assuming data arrives in a streaming fashion. We define the task relationship matrix as *output* kernels in *Reproducing Kernel Hilbert Space* (RKHS) on multitask examples. We propose a novel algorithm called *Online Output Kernel Learning Algorithm* (OOKLA) for lifelong learning setting. For a successful lifelong learning with kernels, we need to address two key challenges: (1) learn the relationships between the tasks (output kernel) efficiently from the data stream and (2) bound the size of the knowledge to avoid memory explosion.

The key challenge in learning with a large number of tasks is to adaptively learn the model parameters and the task relationships, which potentially change over time. Without manageability-efficient updates at each round, learning the task relationship matrix automatically may impose a severe

computational burden. In other words, we need to make predictions and update the models in an efficient real time manner.

We propose simple and quite intuitive update rules for learning the task relationship matrix. When we receive a new example, the algorithm updates the output kernel when the learner made a mistake by computing the similarity between the new example and the set of representative examples (stored in the memory) that belongs to a specific task. If the two examples have similar (different) labels and high similarity, then the relationship between the tasks is increased (decreased) to reflect the positive (negative) correlation and vice versa.

To avoid the memory explosion associated with the lifelong learning setting, we propose a robust budget-limited version of the proposed algorithm that efficiently utilizes the relationship between the tasks to bound the total number of representative examples in the support set. In addition, we propose a two-stage budgeted scheme for efficiently tackling the task-specific budget constraints in lifelong learning.

It is worth noting that the problem of lifelong multitask learning is closely related to online multitask learning. Although the objectives of both online multitask learning and lifelong learning are similar, one key difference is that the online multitask learning, unlike in the lifelong learning, may require that the number of tasks be specified beforehand. In recent years, online multitask learning has attracted extensive research attention Abernethy et al. (2007); Dekel et al. (2007); Lugosi et al. (2009); Cavallanti et al. (2010); Saha et al. (2011); Murugesan et al. (2016). We evaluate our proposed methods with several state-of-the-art online learning algorithms for multiple tasks. Throughout this paper, we refer to our proposed method as online multitask learning or lifelong learning.

There are many useful application areas for lifelong learning, including optimizing financial trading as market conditions evolve, email prioritization with new tasks or preferences emerging, personalized news, and spam filtering, with evolving nature of spam. Consider the latter, where some spam is universal to all users (e.g. financial scams), some messages might be useful to certain affinity groups, but spam to most others (e.g. announcements of meditation classes or other special interest activities), and some may depend on evolving user interests. In spam filtering each user is a "task," and shared interests and dis-interests formulate the inter-task relationship matrix. If we can learn the matrix as well as improving models from specific spam/not-spam decisions, we can perform mass customization of spam filtering, borrowing from spam/not-spam feedback from users with similar preferences. The primary contribution of this paper is precisely the joint learning of inter-task relationships and its use in estimating per-task model parameters in a lifelong learning setting.

## 1.1 RELATED WORK

Most existing work in online learning of multiple task focuses on how to take advantage of task relationships. To achieve this, Lugosi et al. (2009) imposed a hard constraint on the $K$ simultaneous actions taken by the learner in the expert setting, Agarwal et al. (2008) used matrix regularization, and Dekel et al. (2007) proposed a global loss function, as an absolute norm, to tie together the loss values of the individual tasks. Different from existing online multitask learning models, our paper proposes an intuitive and efficient way to learn the task relationship matrix automatically from the data, and to explicitly take into account the learned relationships during model updates.

Cavallanti et al. (2010) assumes that task relationships are available *a priori*. However often such task-relation prior knowledge is either unavailable or infeasible to obtain for many applications especially when the number of tasks $K$ is large (Weinberger et al. (2009)) and/or when the manual annotation of task relationships is expensive (Kshirsagar et al. (2013)). Saha et al. (2011) formulated the learning of task relationship matrix as a Bregman-divergence minimization problem w.r.t. positive definite matrices. The model suffers from high computational complexity as semi-definite programming is required when updating the task relationship matrix at each online round. We show that with a different formulation, we can obtain a similar but much cheaper updating rule for learning the inter-task weights. Murugesan et al. (2016) proposed an efficient method for learning the task relationship matrix using the cross-task performance measure, but their approach learns only the positive correlation between the tasks. Our proposed approach learns positive and negative correlations between the tasks for robust transfer of knowledge from the previously learned tasks.

Recent work in output kernel learning estimate the task covariance matrix in RKHS space, inferred it directly from the data (Dinuzzo et al. (2011); Sindhwani et al. (2013); Jawanpuria et al. (2015)). The task covariance matrix is called the output kernel defined on the tasks, similar to the scalar kernel on the inputs. Most recently, Jawanpuria et al. (2015) showed that for a class of regularization functions, we can efficiently learn this output kernel. Unfortunately most of the proposed methods for learning output kernels require access to the entire data for the learning algorithm, a luxury unavailable in online learning and especially in the lifelong learning setting.

Unlike in online multitask learning, most lifelong learning approaches use a single model for all the tasks or reuse the models from the previous tasks to build a model for the new task (Alquier et al. (2016); Pentina & Urner (2016); Pentina & Ben-David (2015); Balcan et al. (2015); Rusu et al. (2016); Hinton et al. (2015); Rebuffi et al. (2016); Ruvolo & Eaton (2013)). These approaches either increase the computation time on iterations where we encounter a novel task or reduce the prediction power of the model learned from the previous tasks due to catastrophic forgetting. To the best of our knowledge, relationships among the tasks has not been successfully exploited in the lifelong learning setting due to the difficulty in learning a positive semi-definite task relationship matrix in large-scale applications. This paper provides an efficient way to learn the task relationship matrix in the lifelong learning setting.

## 2 LIFELONG LEARNING OF MULTIPLE TASKS

Let $((x_t, i_t), y_t)$ be the example received by the learner from the task $i_t$ (at the time step $t$) where we assume that the $x_t \in \mathcal{X}$ and $y_t$ is its corresponding true label. The task $i_t$ can be a new task or one seen by the learner in the previous iterations. We denote by $[N]$ the consecutive integers ranging from 1 to $N$. In this paper, we do not assume that the number of tasks is known to the learner ahead of time, an important constraint in lifelong learning problems. Let $K$ be the number of tasks seen so far until the current iteration $t$.

For brevity, we consider a binary classification problem for each task $y_t \in \{-1, +1\}$, but the methods generalize to multi-class cases and are also applicable to regression tasks. We assume that the learner made a mistake if $y_t \neq \hat{y}_t$ where $\hat{y}_t$ is the predicted label. Our approach follows a mistake-driven update rule in which the model for a given task is updated only on rounds where the learner predictions differ from the true label.

Let $\mathcal{K} : \mathcal{X} \times \mathcal{X} \to \mathbb{R}$ (kernel on *input* space) and $\Omega : \mathbb{N} \times \mathbb{N} \to \mathbb{R}$ (*output* kernel) be symmetric, positive semi-definite (p.s.d) multitask kernel functions and denote $\mathcal{H}$ as their corresponding RKHS of functions with the norm $\| \cdot \|_{\mathcal{H}}$ on multitask examples (Smola & Schölkopf (1998); Dinuzzo et al. (2011); Jawanpuria et al. (2015)). Using the above notation, we can define a kernel representation of an example based on a set of representative examples collected on the previous iterations (*prototypes*). Formally, given an example $x \in \mathcal{X}$, its kernel representation can be written using this set:

$$x \longmapsto \{\mathcal{K}(x, x_s) : s \in \mathcal{S}\}$$

$\mathcal{S}$ is the set of stored examples for which the learner made a mistake in the past. The set $\mathcal{S}$ is called the *support set*. The online classification function is then defined as the weighted sum of the kernel combination of the examples in the support set. To account for the examples from the different tasks, we consider both the kernel on the input space $\mathcal{K}$ and the output kernel $\Omega$ in our classification function.

$$f(\cdot) = \sum_{(x_s, i_s) \in \mathcal{S}} \alpha_s \Omega_{i_s} \cdot \phi(x_s) \tag{1}$$

We set $\alpha_s = y_s$. The predicted label for a new example is computed from the linear combination of the labels of the examples from the support set $\mathcal{S}$ weighted by their input similarity $\mathcal{K}$ and the task similarity $\Omega$ to the new example. Using the *kernel trick*, one can write:

$$f((x_t, i_t)) = \langle \phi(x_t) \sum_{(x_s, i_s) \in \mathcal{S}} \alpha_s \Omega_{i_s i_t} \phi(x_s) \rangle = \sum_{(x_s, i_s) \in \mathcal{S}} \alpha_s \Omega_{i_s i_t} \mathcal{K}(x_s, x_t)$$

Note that, in the above representation, we need to learn both the support set $\mathcal{S}$ and the output kernel $\Omega$ from the data. As explained in the previous section, for a successful lifelong learning with kernels, we need to address two key challenges: (1) learn the relationships between the tasks (output kernel) efficiently from the data arriving in an online fashion and (2) bound the size of the *support* set $\mathcal{S}$ to avoid memory explosion. We address these two challenges in the following sections.

---

**Algorithm 1:** Online Output Kernel Learning Algorithm

---

**Initialize:** $\mathcal{S} = \emptyset, \mathbf{\Omega}$
**For** $t = 1, 2, \cdots$
    Receive new example $(x_t, i_t)$
    Let $f_{i_t} = \sum_{(x_s, i_s) \in \mathcal{S}} y_s \Omega_{i_s i_t} \phi(x_s)$
    Predict $\hat{y}_t = sign(f_{i_t}(x_t))$
    Receive label $y_t$
    **If** $y_t \neq \hat{y}_t$
        $\mathcal{S} \leftarrow \mathcal{S} \cup (x_t, i_t)$
        Update $\Omega_{i_t k} \forall k \in [K]$

$$\Omega_{i_t k} \leftarrow \Omega_{i_t k} + \frac{1}{\lambda} \sum_{\substack{(x_s, i_s) \in \mathcal{S} \\ i_s = k}} y_t \mathcal{K}(x_t, x_s) y_s \qquad (2)$$

$$\text{(or)}$$

$$\Omega_{i_t k} \leftarrow \Omega_{i_t k} \exp \left\{ \frac{1}{\lambda} \sum_{\substack{(x_s, i_s) \in \mathcal{S} \\ i_s = k}} y_t \mathcal{K}(x_t, x_s) y_s \right\} \qquad (3)$$

    **end**
**end**

---

## 2.1 ONLINE OUTPUT KERNEL LEARNING

Our objective function for the lifelong learning problem is given as follows:

$$\min_{\Omega \in S_+, f \in \mathcal{H}} \sum_t \ell(y_t, f((x_t, i_t))) + \frac{1}{2} \|f\|_{\mathcal{H}}^2 + \lambda \mathcal{R}(\Omega) \qquad (4)$$

where $\ell(\cdot)$ is some loss function such as hinge loss or logistic loss, $\mathcal{R}(\cdot)$ is the regularization on the task relationship matrix $\Omega$ and $\lambda$ is the regularization parameter. Note that $f$ in the above equation depends on $\Omega$. In order to reduce the time taken for each time-step, we require an efficient update to the task relationship matrix $\Omega$. Following the work of Jawanpuria et al. (2015) in the batch setting, we consider a subset of regularization functions $\mathcal{R}$ for which we can efficiently learn the task covariance matrix. Consider the dual function of the above equation, at time-step $t$ (see Barzilai & Crammer (2015); Jawanpuria et al. (2015)):

$$\max_{\Omega \in S_+} \frac{1}{2} \sum_k \sum_{\substack{(x_s, i_s) \in \mathcal{S} \\ i_s = k}} \alpha_t \mathcal{K}(x_t, x_s) \alpha_s \Omega_{i_t k} - \lambda \mathcal{R}(\Omega, \Omega_{(t-1)}) \qquad (5)$$

When we consider the entry-wise $l_p$ norm between $\Omega$ and $\Omega_{(t-1)}$ from the previous iteration as our regularization i.e., $\mathcal{R}(\Omega, \Omega_{(t-1)}) = \sum_k |\Omega_{i_t k} - (\Omega_{(t-1)})_{i_t k}|^p$ for $p \geq 1$ [1], we get the update function in Equation 2. Similarly, if we consider the generalized KL-divergence between $\Omega$ and $\Omega_{(t-1)}$ i.e., $\mathcal{R}(\Omega, \Omega_{(t-1)}) = \sum_k \Omega_{i_t k} \log \frac{\Omega_{i_t k}}{(\Omega_{(t-1)})_{i_t k}} - \Omega_{i_t k} + (\Omega_{(t-1)})_{i_t k}$, we get the update function in Equation 3. Unlike in the previous work, we update only the row (and the corresponding column) of the task relationship matrix $\Omega$ specific to the task $i_t$, which significantly reduces the time taken per example.

We can see that the update equations are simple and quite intuitive. For a given new example $(x_t, i_t)$ at round $t$, the algorithm updates $\Omega_{i_t k}$ (for some $k \in [K]$) by computing the similarity between the new example and the examples in the support set $\mathcal{S}$ that belongs to the task $k$. If the two examples have similar (different) labels and high similarity $\mathcal{K}(x_t, x_s)$, then the $\Omega_{i_t k}$ is increased to reflect the positive (negative) correlation and vice versa. A value close to 0 implies no significant relationship between the tasks. The update to the $\Omega_{i_t k}$ is normalized by the regularization parameter $\lambda$ for scaling.

---

[1] We set $p = 2$ in this paper.

It is worth noting that our update equations do not violate the *p.s.d* constraints on $\Omega$ in Equation 5. If $\Omega$ from the previous iteration is a *p.s.d* matrix and the update is a *p.s.d* matrix (as it is computed using the Gram matrix of the example from the previous iteration), the sum and Hadamard product of two *p.s.d* matrices satisfy the *p.s.d* constraint (using the Schur Product Theorem).

Algorithm 1 outlines the key steps in our proposed method. We write $f((x_t, i_t))$ as $f_{i_t}(x_t)$ for notational convenience. At each time-step $t$, the learner *receives* an example $x_t \in \mathcal{X}$ and *predicts* the output label $y_t$ using $\hat{y}_t = sign(f_{i_t}(x_t))$. We update both the support set $\mathcal{S}$ and the output kernel $\Omega_{i_t}$. when the learner makes a mistake.

---

**Algorithm 2:** Budgeted OOKLA

**Initialize:** $\mathcal{S} = \emptyset$, $\mathbf{\Omega}$
**For** $t = 1, 2, \cdots$
    Receive new example $(x_t, i_t)$
    Let $f_{i_t} = \sum_{(x_s, i_s) \in \mathcal{S}} y_s \Omega_{i_s i_t} \phi(x_s)$
    Predict $\hat{y}_t = sign(f_{i_t}(x_t))$
    Receive label $y_t$
    **If** $y_t \neq \hat{y}_t$
        **If** $|\mathcal{S}| < B$
            $\mathcal{S} \leftarrow \mathcal{S} \cup (x_t, i_t)$
        **Else**
            Find an example to remove

$$\arg\max_{(x_r, i_r) \in \mathcal{S}} \sum_{k \in [K]} \Omega_{i_r k} \big[ y_r \big( f_k - y_r \Omega_{i_r k} \phi(x_r) \big) \phi(x_r) \big] \quad (6)$$

            $\mathcal{S} \leftarrow \mathcal{S} \cup (x_t, i_t) \setminus (x_r, i_r)$
        **end**
        Update $\Omega_{i_t k} \forall k \in [K]$
        as in Algorithm 1.
    **end**
**end**

---

**Algorithm 3:** Two-Stage Budgeted Learning

**Initialize:** $\mathcal{S} = \emptyset$, $\mathcal{T}_k = \emptyset$, $k \in [K]$, $\mathbf{\Omega}$
**For** $t = 1, 2, \cdots$
    Receive new example $(x_t, i_t)$
    Let $f_{i_t} = \sum_{(x_s, i_s) \in \mathcal{T}_{i_t}} y_s \Omega_{i_s i_t} \phi(x_s)$
    Predict $\hat{y}_t = sign(f_{i_t}(x_t))$
    Receive label $y_t$
    **If** $y_t \neq \hat{y}_t$
        $\mathcal{S} \leftarrow \mathcal{S} \cup \{t\} \setminus \{r\}$
        `/* r=∅ if |S| < B, else`
        `    choose r using Equation 6`
        `*/`
        **For** $k \in [K]$
            $\mathcal{T}_k \leftarrow \mathcal{T}_k \cup \{l\} \setminus \{r\}$
            `/* remove r from T_k;`
            `    choose l from S − T_k   */`
        **end**
        Update $\Omega_{i_t k} \forall k \in [K]$ as in Algorithm 1.
    **end**
**end**

---

## 2.2 BUDGETED LEARNING

In Algorithm 1, we can see that both the classification function $f$ and the update equations for $\Omega$ use the support set $\mathcal{S}$. When the target function changes over time, the support set $\mathcal{S}$ grows unboundedly. This leads to serious computational and runtime issues especially in the lifelong learning setting. The most common solution to this problem is to impose a bound on the number of examples in the support set $\mathcal{S}$. There are several budget maintenance strategies proposed recently (Cavallanti et al. (2007); Dekel et al. (2008); Orabona et al. (2008)). Unfortunately these schemes cannot be directly used in our setting due to the output kernels in our learning formulation. Cavallanti & Cesa-Bianchi (2012) proposed multitask variants of these schemes but they are impractical for the lifelong learning setting. We follow a simple support set removal schemes based on Crammer et al. (2004). In single-task setting, when the number of examples in the support set $\mathcal{S}$ exceeds the limit (say $B$), a simple removal scheme chooses an example $x_r$ with the highest confidence from $\mathcal{S}$. The confidence of an example $x_r$ is measured using $y_r f(x_r)$ after removing $x_r$ from the support set $\mathcal{S}$.

$$\arg\max_{x_r \in \mathcal{S}} \big[ y_r \big( f - y_r \phi(x_r) \big) \phi(x_r) \big] \quad (7)$$

We extend the above approach to the multitask and lifelong learning settings. Since the support set $\mathcal{S}$ is shared by all the tasks, we choose an example $x_r$ with high confidence to remove from each task function $f_k$, weighted by the relationship among the tasks. The objective function to choose the example is shown in Equation 6. We show in the experiment section that this simple approach is efficient and performs significantly better than the state-of-the-art budget maintenance strategies. Algorithm 2 shows pseudocode of the proposed budgeted learning algorithm.

### 2.2.1 TWO-STAGE BUDGETED LEARNING

In lifelong learning setting, the number of tasks is typically large. The support set $\mathcal{S}$ may have hundreds or thousands of examples from all the tasks. Each task does not use all the examples from the support set $\mathcal{S}$. For example, in movie recommendations task, recommendation for each user (task) can be characterized by just a few movies (subset of examples) in the support set $\mathcal{S}$. Motivated by this observation, we propose a two-stage budgeted learning algorithm for the lifelong learning setting.

Algorithm 3 shows pseudocode of the proposed two-stage budgeted learning algorithm. In addition to the support set $\mathcal{S}$, we maintain task-specific support set $\mathcal{T}_k$. We choose the budget for each task (say $L$) where $L <<< B$. Similar to the removal strategies for $\mathcal{S}$, we remove an example from $\mathcal{T}_k$ when $|\mathcal{T}_k| > L$ and replace with an example from the set $\mathcal{S} - \mathcal{T}_k$. The proposed two-stage approach provides better runtime complexity compared to the budgeted algorithm in Algorithm 2. Since only a subset of tasks may hold an example from $\mathcal{S}$, the removal step in Equation 6 requires only a subset of tasks for choosing an example. This improves the runtime per iteration significantly when the number of tasks is large. One may consider a different budget size for each task $L_k$ based on the complexity of the task.

In addition, the proposed two-stage budgeted learning algorithm provides an alternative approach to using state-of-the-art budget maintenance strategies. For example, it is easier to use the Projectron algorithm (Orabona et al. (2008)) on $\mathcal{T}_k$, rather than on $\mathcal{S}$. We will further explore this line of research in our future work.

## 3 EXPERIMENTS

In this section, we evaluate the performance of our algorithms. All reported results are averaged over 10 random runs on permutations of the training data. Unless otherwise specified, all model parameters are chosen via 5-fold cross validation.

### 3.1 BENCHMARK DATASETS

We use three benchmark datasets, commonly used for evaluating online multitask learning. Details are given below:

**Newsgroups Dataset**[2] consists of 20 tasks generated from two subject groups: *comp* and *talk.politics*. We paired two newsgroups, one from each subject (e.g.,comp.graphics vs talk.politics.guns), for each task. In order to account for positive/negative correlation between the tasks, we randomly choose one of the newsgroups as positive $(+)$ or negative $(-)$ class. Each post in a newsgroup is represented by a vocabulary of approximately $60K$ unique features.

**Spam Dataset**[3] We use the dataset obtained from ECML PAKDD 2006 Discovery challenge for the spam detection task. We used the task B challenge dataset, which consists of labeled training data from the inboxes of 15 users. We consider each user as a single task and the goal is to build a personalized spam filter for each user. Each task is a binary classification problem: spam $(+)$ or non-spam $(-)$ and each example consists of approximately $150K$ features representing term frequency of the word occurrences. Some spam is universal to all users (e.g. financial scams), but some messages might be useful to certain affinity groups and spam to most others. Such adaptive behavior of each user's interests and dis-interests can be modeled efficiently by utilizing the data from other users to learn per-user model parameters.

**Sentiment Dataset**[4] We also evaluated our algorithm on product reviews from amazon. The dataset contains product reviews from 25 domains. We consider each domain as a binary classification task. Reviews with rating > 3 were labeled positive $(+)$, those with rating < 3 were labeled negative $(-)$, reviews with rating = 3 are discarded as the sentiments were ambiguous and hard to predict. Similar to the previous datasets, each example consists of approximately $350K$ features representing term frequency of the word occurrences.

---

[2] http://qwone.com/~jason/20Newsgroups/
[3] http://ecmlpkdd2006.org/challenge.html
[4] http://www.cs.jhu.edu/~mdredze/datasets/sentiment

Table 1: Average performance on three datasets: means and standard (test set) scores over 10 random shuffles.

| Models | Newsgroups | | | Spam | | | Sentiment | | |
|---|---|---|---|---|---|---|---|---|---|
| | AUC | nSV | Time (s) | AUC | nSV | Time (s) | AUC | nSV | Time (s) |
| Perceptron | 0.7974 (0.01) | 1421.6 | 8.36 | 0.8621 (0.01) | 1110.0 | 9.23 | 0.6363 (0.01) | 1018.8 | 255.3 |
| PA | 0.7633 (0.02) | 1926.6 | 9.89 | 0.8394 (0.01) | 1482.1 | 11.5 | 0.5321 (0.01) | 2500.0 | 357.7 |
| FOML | 0.7830 (0.01) | 1592.0 | 18.46 | **0.8803** (0.01) | 1228.2 | 16.0 | 0.6484 (0.01) | 1030.0 | 266.0 |
| OMTRL | 0.7694 (0.01) | 1560.4 | 72.36 | 0.8583 (0.01) | 1266.3 | 45.4 | 0.5651 (0.01) | 1057.6 | 328.2 |
| OSMTL | 0.7969 (0.02) | 1426.2 | 18.18 | 0.8611 (0.01) | 1115.1 | 17.3 | 0.6425 (0.01) | 1021.8 | 264.3 |
| OOKLA-*sum* | **0.8512** (0.02) | 884.9 | 15.58 | **0.8825** (0.01) | 833.4 | 17.3 | 0.6261 (0.01) | 925.0 | 306.6 |
| OOKLA-*exp* | 0.8382 (0.02) | 886.9 | 15.50 | **0.8819** (0.01) | 1062.5 | 19.2 | **0.6731** (0.01) | 882.2 | 302.8 |

We choose 2000 examples (100 posts per task) for *20 Newsgroups*, 1500 emails for *spam* (100 emails per user inbox) and 2500 reviews for *sentiment* (100 reviews per domain) as training set for our experiments. Note that we intentionally kept the size of the training data small to simulate the lifelong learning setting and drive the need for learning from previous tasks, which diminishes as the training sets per task become large. Since these datasets have a class-imbalance issue (with few $(+)$ examples as compared to $(-)$ examples), we use average Area Under the ROC Curve ($AUC$) as the performance measure on the test set.

## 3.2 RESULTS

To evaluate the performance of our proposed algorithm (*OOKLA*), we use the three datasets (*Newsgroups*, *Spam* and *Sentiment*) for evaluation and compare our proposed methods to 5 baselines. We implemented Perceptron and Passive-Aggressive algorithm (*PA*) Crammer et al. (2006) for online multitask learning. Both Perceptron and *PA* learn independent model for each task. These two baselines do not exploit the task-relationship or the data from other tasks during model update. Next, we implemented two online multitask learning related to our approach: *FOML* – initializes $\Omega$ with fixed weights Cavallanti et al. (2010), Online Multitask Relationship Learning (*OMTRL*) Saha et al. (2011) – learns a task covariance matrix along with task parameters. Since *OMTRL* requires expensive calls to SVD routines, we update the task-relationship matrix every 10 iterations. In addition, we compare our proposed methods against the performance of Online Smooth Multitask Learning (*OSMTL*) which learns a probabilistic distribution over all tasks, and adaptively refines the distribution over time Murugesan et al. (2016). We implement two versions of our proposed algorithm with different update rules for the task-relationship matrix: *OOKLA-sum* (Equation 2 OOKLA with sum update) *OOKLA-exp* (Equation 3 OOKLA with exponential update) as shown in Algorithm 1.

Table 1 summarizes the performance of all the above algorithms on the three datasets. In addition to the $AUC$ scores, we report the average total number of support vectors (*nSV*) and the CPU time taken for learning from one instance (*Time*).

From the table, it is evident that both OOKLA-*sum* and OOKLA-*exp* outperform all the baselines in terms of both $AUC$ and *nSV*. This is expected for the two default baselines (*Perceptron* and *PA*). The update rule for *FOML* is similar to ours but using fixed weights. The results justify our claim that learning the task-relationship matrix adaptively leads to improved performance. As expected, both *OOKLA* and *OSMTL* consume less or comparable CPU time than the subset of baselines which take into account learning inter-task relationships. Unlike in the *OMTRL* algorithm that recomputes the task covariance matrix every (10) iteration using expensive SVD routines, the task-relationship matrix in our proposed methods (and *OSMTL*) are updated independently for each task. We implement the *OSMTL* with exponential update for our experiments as it has shown to perform better than the other baselines. One of the major drawbacks of *OSMTL* is that it learn only the positive correlations between the tasks. The performance of *OSMTL* worsens when the tasks are negatively correlated. As

Table 2: Evaluation of proposed methods with different budgets: means and standard AUC (test set) scores over 10 random shuffles.

| Models | Newsgroups | | | Spam | | | Sentiment | | |
|---|---|---|---|---|---|---|---|---|---|
| Budget *(B)* | *50* | *100* | *150* | *50* | *100* | *150* | *50* | *100* | *150* |
| RBP | 0.6272 | 0.6954 | 0.7373 | 0.6426 | 0.7197 | 0.7536 | 0.5679 | 0.5790 | 0.5889 |
| | (0.07) | (0.03) | (0.01) | (0.06) | (0.03) | (0.04) | (0.02) | (0.02) | (0.02) |
| | 6.5s | 6.4s | 7.4s | 14.5s | 18.5s | 16.3s | 1772.4s | 1666.9s | 1860.2s |
| Forgetron | 0.6353 | 0.6860 | 0.7240 | 0.6546 | 0.7403 | 0.7339 | 0.5579 | 0.5847 | 0.6195 |
| | (0.07) | (0.05) | (0.01) | (0.06) | (0.04) | (0.04) | (0.02) | (0.02) | (0.02) |
| | 9.1s | 8.6s | 9.9s | 20.5s | 21.4s | 21.9s | 2219.1s | 2097.0s | 2272.6s |
| Projectron | 0.6399 | 0.6984 | 0.6986 | 0.6352 | 0.6041 | 0.6402 | 0.5215 | 0.5187 | 0.5221 |
| | (0.03) | (0.02) | (0.02) | (0.03) | (0.05) | (0.05) | (0.01) | (0.01) | (0.02) |
| | 16.7s | 16.6s | 17.6s | 24.7s | 26.1s | 27.9s | 2712.0s | 2622.5s | 2804.5s |
| Budget -OKL | 0.6810 | 0.7106 | 0.7499 | 0.6472 | 0.6471 | 0.6552 | 0.5683 | 0.5732 | 0.6153 |
| | (0.02) | (0.03) | (0.03) | (0.03) | (0.03) | (0.04) | (0.02) | (0.01) | (0.01) |
| | 6.3s | 10.2s | 13.4s | 18.8s | 22.6s | 24.7s | 1708.8s | 1674.4s | 1935.2s |
| Budget -OOKLA | **0.7576** | **0.7562** | **0.7749** | **0.7485** | **0.7969** | **0.8472** | **0.6117** | **0.6901** | **0.7000** |
| | (0.04) | (0.04) | (0.03) | (0.08) | (0.04) | (0.02) | (0.06) | (0.01) | (0.04) |
| | 9.9s | 11.7s | 14.1s | 20.1s | 24.3s | 26.9s | 2250.5s | 2073.0s | 2334.3.3s |

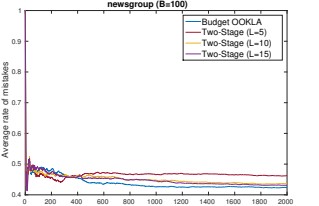 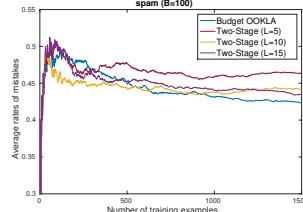 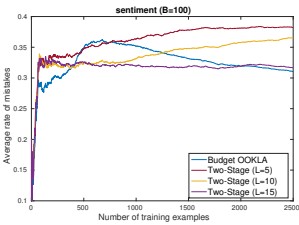

Figure 1: Number of training examples vs Average rate of mistakes calculated for different values of $L$.

we can see from the table, our proposed methods outperform *OSMTL* significantly in the *Newsgroup* dataset.

Table 2 compares the proposed methods with different budget schemes and budget sizes in terms of test set $AUC$ scores and the runtime. We use *OOKLA-sum* for this experiment. We set the value of $B$ to $\{50, 100, 150\}$ for all the datasets. We compare our proposed budgeted learning algorithm (Algorithm 2) with the following state-of-the-art algorithms for online budgeted learning: (1) Random Budgeted Perceptron (RBP) Cavallanti et al. (2007)- randomly chooses an example to remove (2) (Self-tuned) Forgetron Dekel et al. (2008)- forgets the oldest example (3) Projectron++ Orabona et al. (2008)- projects the new example on the others (4) Budgeted Online Kernel learning (Budget-OKL)- removes an example using equation 7 (5) Budgeted Online Output Kernel Learning (Budget-OOKLA)- removes an example using equation 6. Following Cavallanti & Cesa-Bianchi (2012), we implement the online multitask version of RBP, Forgetron and Projectron for our experiments.

Table 2 shows both the test set AUC scores (first line) and time taken for learning from one instance (including the removal step). It is evident from the table, our proposed budgeted learning algorithm for online multitask learning significantly outperforms the other state-of-the-art budget schemes on most settings. Our proposed algorithm uses the relationship between the tasks efficiently to choose the next example for removal.

Finally, we evaluate the performance of the proposed two-stage budgeted scheme compared to the Algorithm 2. To study the effect of different budget sizes $L$, we compute the cumulative mistake rate $\frac{\sum_t \mathbb{I}\{y_t \neq \hat{y}_t\}}{t}$ measured at each time-step $t$. Figures 1 show average rate of mistakes against the number of training examples seen so far on *newsgroup*, *spam* and *sentiment* datasets for different values of $L$. We fix the value of $B = 100$. As we can see from the figures, even for small values of $L$, the average mistake rate of the two-stage budgeted learning algorithm is comparable to the model

which uses all the examples from the support set $\mathcal{S}$. We observe similar trend in the test set AUC scores. On average, we achieved over $16\%$ improvement in running time compared to the budget maintenance scheme in Algorithm 2. We believe that the time consumption and the performance improvement will be even better for applications with larger numbers of tasks.

## 4  CONCLUSIONS

We proposed a novel lifelong learning algorithm using output kernels. The proposed method efficiently learns both the model and the inter-task relationships at each iteration. Our update rules for learning the task relationship matrix, at each iteration, were motivated by the recent work in output kernel learning.

In order to handle the memory explosion from an unbounded support set in the lifelong learning setting, we proposed a new budget maintenance scheme that utilizes the task relationship matrix to remove the least-useful (high confidence) example from the support set. In addition, we proposed a two-stage budget learning scheme based on the intuition that each task only requires a subset of the representative examples in the support set for efficient learning. It provides a competitive and efficient approach to handle large number of tasks in many real-life applications.

The effectiveness of our algorithm is empirically verified over several benchmark datasets, outperforming several competitive baselines both in the unconstrained case and the budget-limited case, where selective forgetting was required.

Future research includes other forms of budget limits (e.g. time or reprocessing capacity), more complex inter-task relationships (e.g. hierarchical), and additional factors such as task reliability (e.g. prefer transferring from a well-trained related task, vs from a relatively data-sparse one, whose model is less reliable).

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
