# OpenReview forum: "Lifelong Learning with Output Kernels"
_ICLR.cc/2018/Conference — Reject_

### Official Review · AnonReviewer2 · 2017-11-22
**This paper addresses the problem of lifelong multitask learning and proposes an efficient updating rule to learning the inter-task relationship and a new budget maintenance scheme to overcome the out of memory issue.  Experiments are conducted to demonstrate the effectiveness of the proposed method in the limited budget situation.**

**Rating:** 3
**Confidence:** 4

**Review:**

CONTRIBUTION
The main contribution of the paper is not clearly stated.  To the reviewer, It seems “life-long learning” is the same as “online learning”.  However, the whole paper does not define what “life-long learning” is.
The limited budget scheme is well established in the literature.
1. J. Hu, H. Yang, I. King, M. R. Lyu, and A. M.-C. So. Kernelized online imbalanced learning with fixed budgets. In AAAI, Austin Texas, USA, Jan. 25-30 2015.
2. Y. Engel, S. Mannor, and R. Meir. The kernel recursive least-squares algorithm. IEEE Transactions on Signal Processing, 52(8):2275–2285, 2004.
It is not clear what the new proposal in the paper.

WRITING QUALITY
The paper is not well written in a good shape. Many meanings of the equations are not stated clearly, e.g., $phi$ in eq. (7). Furthermore, the equation in algorithm 2 is not well formatted.

DETAILED COMMENTS
1. The mapping function $phi$ appears in Eq. (1) without definition.
2. The last equation in pp. 3 defines the decision function f by an inner product. In the equation, the notation x_t and i_t is not clearly defined.  More seriously, a comma is missed in the definition of the inner product.
3. Some equations are labeled but never referenced, e.g., Eq. (4).
4. The physical meaning of Eq.(7) is unclear.  However, this equation is the key proposal of the paper.   For example, what is the output of the Eq. (7)? What is the main objective of Eq. (7)?  Moreover, what support vectors should be removed by optimizing Eq. (7)?  One main issue is that the notation $phi$ is not clearly defined.   The computation of f-y_r\phi(s_r) makes it hard to understand.  Especially,  the dimension of $phi$ in Eq.(7) is unknown.

ABOUT EXPERIMENTS
1.	It is unclear how to tune the hyperparameters.
2.	In Table 1, the results only report the standard deviation of AUC. No standard deviations of nSV and Time are reported.

---

> ### Author Response · Authors · 2018-01-06
> **@AnonReviewer2**
>
> “Difference between online multitask learning and lifelong learning”
> As mentioned in Page 2 Paragraph 3, the key difference is that the online multitask learning, unlike in the lifelong learning, may require that the number of tasks be specified beforehand. Most existing online multitask learning algorithms utilize this additional knowledge for learning the task relationship such as FOML, OMTRL, OSMTL, etc.
>
> “References for budgeted schemes”
> Thank you for the additional references. The budget schemes from Hu et al. use the similarity between the examples for removal, on the other hand, our proposed budget schemes consider both the similarity between the examples and the relationship between the tasks to identify an example to remove.  The sparsification procedure considered in Engel et al.  will suffer from scalability issues similar to the Multitask Projectron as discussed in the paper.
>
> \phi(.) is the feature function used in kernel learning. We will make this clear with all other clarifications in our revised version.

---

### Official Review · AnonReviewer1 · 2017-11-25
**A lifelong mutlitask learning scheme with online kernel learning, with low technical novelty**

**Rating:** 2
**Confidence:** 5

**Review:**

Summary: The paper proposed a two-dimensional approach to lifelong learning, in the context of multi-task learning. It receives instances in an online setting, where both the prediction model and the relationship between the tasks are learnt using a online kernel based approach. It also proposed to use budgeting techniques to overcome computational costs. In general, the paper is poorly written, with many notation mistakes and inconsistencies. The idea does not seem to be novel, technical novelty is low, and the execution in experiments does not seem to be reliable.

Quality: No obvious mistakes in the proposed method, but has very low novelty (as most methods follows existing studies in especially for online kernel learning). Many mistakes in the presentation and experiments.

Originality: The ideas do not seem to be novel, and are mostly (trivially) using existing work as different components of the proposed technique.

Clarity: The paper makes many mistakes, and is difficult to read. [N] is elsewhere denoted as \mathbb{N}. The main equation of Algorithm 2 merges into Algorithm 3. Many claims are made without justification (e.g. 2.2. “Cavallanti 2012 is not suitable for lifelong learning”… why?; “simple removal scheme … highest confidence” – what is the meaning of highest confidence?), etc. The removal strategy is not at all well explained – the objective function details and solving it are not discussed.

Significance: There is no theoretical guarantee on the performance, despite the author’s claiming this as a goal in the introduction itself (“goal of lifelong learner … computation”). The experiments are not reliable. Perceptron obtains a better performance than PA algorithms – which is very odd. Moreover, many of the multi-task baselines obtain a worse performance than a simple perceptron (which does not account for multi-task relationships).

---

> ### Author Response · Authors · 2018-01-06
> **@AnonReviewer1**
>
> “Cavallanti 2012 is impractical for lifelong learning”
> The multitask variants of the budgeted schemes in Cavallanti 2012 assumes that the relationship between the tasks are known a priori. In addition to the unknown number of tasks, they don’t scale to lifelong learning setting since the tasks arrive sequentially.
>
> “Confidence in removal step”
> The confidence is measured using the margin i.e., how far an example x_r is from the margin after removing it from S.
>
> “multitask baselines worse than perceptron”
> Perceptron in Table 1 shows the results for single-task setting where it builds one models for all the tasks, whereas PA shows the results for independent task learning where it learns independent model for each task.
>
> Since Perceptron, FOML and OSMTL cannot learn the negative correlation between the tasks, the results of Perceptron, FOML and OSMTL are similar in newsgroup datasets.  Note that the results for Perceptron is comparable to that of FOML and OSMTL as it sets \Omega_{ij}=1 for all i,j.  In case of sentiment dataset, we can see that FOML and OSMTL outperform Perceptron as they consider the task relationship. We will fix this in the revised version.

---

### Official Review · AnonReviewer3 · 2017-11-28
**The paper in an online budgeted version of an existing lifelong learning algorithm. The methodological contribution is minor and the experiments are not well designed.**

**Rating:** 4
**Confidence:** 4

**Review:**

The paper proposes a budgeted online kernel algorithm for multi-task learning. The main contribution of the paper is an online update of the output kernel, which measures similarity between pairs of tasks. The paper also proposes a removal strategy that bounds the number of support vectors in the kernel machine. The proposed algorithm is tested on 3 data sets and compared with several baselines.
  Positives:
- the output kernel update is well justified
- experimental results are encouraging
  Negatives:
- the methodological contribution of the paper is minimal
- the proposed approach to maintain the budget is simplistic
- no theoretical analysis of the proposed algorithm is provided
- there are issues with the experiments: the choice of data sets is questionable (all data sets are very small so there is not need for online learning or budgeting; newsgroups is a multi-class problem, so we would want to see comparisons with some good multi-class algorithms; spam data set might be too small), it is not clear what were hyperparameters in different algorithms and how they were selected, the budgeted baselines used in the experiments  are not state of the art (forgetron and random removal are known to perform poorly in practice, projectron usually works much better), it is not clear how a practitioner would decide whether to use update (2) or(3)

---

> ### Author Response · Authors · 2018-01-06
> **@AnonReviewer3**
>
> “all data sets are very small so”
> *) “there is no need for online learning”
> Our focus in this paper is on the scenario where training examples are insufficient for each single task.  In other words, we are interested in a lifelong learning setting where we see large number of tasks with limited set of labeled examples per task.
>
> *) “or budgeting”
> The budget/support set S contains examples from all the tasks. Even though the number of examples per task is small, the tasks arrive sequentially (with unknown horizon). The new examples are added to S over several rounds. Without any bound on the size of S, we will face the memory explosion problem as discussed in the Introduction section.
>
> “newsgroups is a multi-class problem”
> We use newsgroup dataset to demonstrate the effectiveness of the proposed algorithm to learn the (positive, negative, no) correlation between the tasks with our simple update rules. In this experiment, each task identifies the subject group (comp and talk.politics) rather than the class.
>
> “budgeted baselines are not state-of-the-art”
> Our baselines in Table 2 are specific to multitask and lifelong learning setting that considers relationship between the tasks for removal step (See Cavallanti 2012). In addition, Projectron has one of the best retention policies for budgeted learning algorithms.

---

### Author Response · Authors · 2018-01-06
**Major Clarification Points**

We thank all the reviewers for the helpful comments, which we will take into account in our revision.  We give our major clarifications points here.

 “Minimal contribution”:
In essence, our paper has three main contributions:
1) proposing simple update rules for learning task relationship
Unlike in (Jawanpuria 2015), we proposed simple sequential updates for learning task relationship matrix in addition to satisfying its positive semi-definite constraint at each iteration.  These update equations are unique to online lifelong learning setting and doesn’t require access to the entire input kernel matrix. The proposed algorithm can easily scales to large datasets with many tasks.

2) incorporating task relationship in the budgeted scheme
Our proposed scheme consists of a simple removal step that utilizes the task relationship.  Unlike in the previous work, we remove an example from the support set S by considering both the similarity between the examples (via confidence of the models) and the relationship between the tasks with less runtime per removal step. The proposed method empirically outperforms (both in terms of AUC and Time taken) other multitask budgeted learning schemes.

3) two-stage budgeted approach for lifelong learning
To the best of our knowledge, our paper proposed the first practical budgeted scheme for lifelong multitask learning. The two-stage budgeted scheme allows us to use expensive budgeted schemes with best retention policy such as Projectron on task-specific budget T_k instead of S.

We will clarify these points along with additional details on update equations and budgeted removal step.

“Smaller datasets”
All the datasets in our experiments are widely accepted benchmarks in online multi-task and lifelong learning evaluations (See Pentina 2016, Murugesan 2016).  We chose these 3 datasets for two main reasons: 1) for a fair comparison with the current online multi-task learning methods such as OSMTL, OMTRL, etc. 2) to consider different type of tasks that one may encounter in many practical applications such as spam detection, sentiment analysis, etc.

We plan to include additional experiments on datasets with large number of tasks in the revised version.

“Hyper-parameters”
We tuned all the hyper-parameters via 5-fold cross validation. We will include additional details on the hyper-parameters of the baselines for clarity.

“Theoretical analysis”
We are currently working on the theoretical bounds for the proposed lifelong learning approach. We will derive the generalization bounds for lifelong learning setting with respect to some unknown task-generating probability distribution.

---

### Decision · Program_Chairs · 2018-01-29
**ICLR 2018 Conference Acceptance Decision**

**Decision:**

Reject

**Comment:**

The output kernel idea for lifelong learning is interesting, but insufficiently developed in the current draft.